

# Synergy between satellite observations and model simulations during extreme events

Anne Wiese[1], Joanna Staneva[1], Johannes Schultz-Stellenfleth[1], Arno Behrens[1], Luciana Fenoglio-Marc[2], and Jean-Raymond Bidlot[3]

[1]Institute of Coastal Research, Helmholtz-Zentrum Geesthacht, Geesthacht, Germany
[2]Institute of Geodesy and Geoinformation, University of Bonn, Bonn, Germany
[3]European Centre for Medium-Range Weather Forecasts, Reading, UK

**Correspondence:** Anne Wiese (anne.wiese@hzg.de)

**Abstract.** In this study, the quality of wind and wave data provided by the new Sentinel-3A satellite is evaluated. We focus on coastal areas, where altimeter data are of lower quality than those for the open ocean. The satellite data of Sentinel-3A, Jason-2 and CryoSat-2 are assessed in a comparison with in situ measurements and spectral wave model (WAM) simulations. The sensitivity of the wave model to wind forcing is evaluated using data with different temporal and spatial resolution, such as

ERA-Interim and ERA5 reanalyses, ECMWF operational analysis and short-range forecasts, German Weather Service (DWD) forecasts and regional atmospheric model simulations -coastDat. Numerical simulations show that both the wave model forced using the ERA5 reanalyses and that forced using the ECMWF operational analysis/forecast demonstrate the best capability over the whole study period, as well as during extreme events. To further estimate the variance of the significant wave height of ensemble members for different wind forcings, especially during extreme events, an empirical orthogonal function (EOF)

analysis is performed. Intercomparisons between remote sensing and in situ observations demonstrate that the overall quality of the former is good over the North Sea and Baltic Sea throughout the study period, although the significant wave heights estimated based on satellite data tend to be greater than the in situ measurements by 7 cm to 26 cm. The quality of all satellite data near the coastal area decreases; however, within 10 km off the coast, Sentinel-3A performs better than the other two satellites. Analyses in which data from satellite tracks are separated in terms of onshore and offshore flights have been

carried out. No substantial differences are found when comparing the statistics for onshore and offshore flights. Moreover, no substantial differences are found between satellite tracks under various metocean conditions. Furthermore, the satellite data quality does not depend on the wind direction relative to the flight direction. Thus, the quality of the data obtained by the new Sentinel-3A satellite over coastal areas is improved compared to that of older satellites.

## 1 Introduction

Information on the state of the sea in coastal areas is of great interest, as waves are a crucial factor of important activities conducted at sea. Therefore, an accurate wave forecast and hindcast is very important for marine traffic, recreational activities on the water, urban development near the coast, ecosystem restoration, renewable energies and offshore management (Gautier and Caires, 2015; Thomas and Dwarakish, 2015). Global ocean wave forecasts with coarser spatial resolution have already reached



a remarkable level of accuracy (Janssen and Bidlot, 2018). However, for inner basins and coastal areas, higher resolution is required, and numerical wave models still have some deficits (Cavaleri and Bertotti, 2003b; Van Vledder and Akpınar, 2015).

In many studies, the meteorological input has already been found to be a crucial factor for conducting good wave forecasts (Teixeira et al., 1995; Cavaleri and Bertotti, 2003b, 2004; Cavaleri et al., 2007; Thomas and Dwarakish, 2015; Van Vledder and
Akpınar, 2015). The wind data used to force a wave model needs to be very accurate since, in coastal areas, the fetch is limited and small islands can block wave propagation. Small changes in wind direction can lead to drastically different wave results. The wind speed is a crucial factor of the significant wave height. However, peaks and extreme events are frequently not well simulated by the wave model because the meteorological input underestimates the wind speed (Cavaleri et al., 2007; Cavaleri, 2009). More than 20 years ago, Cavaleri and Bertotti (1997) suggested that the general performance of the wave model as well
as its performance during extreme events can be improved by using a wind input field with a higher spatial resolution. Since the most advanced wave models at that time were more accurate than the meteorological ones, the quality of the wave model output was a very good indicator of the quality of the meteorological input data. Cavaleri and Bertotti (2003b, 2004) analysed the accuracy of the modelled wind and wave fields of enclosed seas, such as the Mediterranean Sea, with respect to the spatial resolution of the wind fields. They found that the modelled surface wind speeds are almost always underestimated, which they
attributed to a lack of spatial resolution (Cavaleri and Bertotti, 2003b). When the meteorological input data has a higher spatial resolution, the average results of the wave model are indeed closer to the ground truth (Cavaleri and Bertotti, 2004). However, even today, wind data inaccuracy leads to discrepancies between wave model simulations (Thomas and Dwarakish, 2015; Van Vledder and Akpınar, 2015). Van Vledder and Akpınar (2015) assessed the sensitivity of the wave model SWAN to the spatial and temporal resolution of wind input data in the area of the Black Sea. They concluded that the wave model results are
critically sensitive to the spatial resolution and less sensitive to the temporal resolution of the meteorological input data. Similar analyses have been conducted both globally (Feng et al., 2006) and for coastal areas such as that around the Mediterranean Sea (Cavaleri and Bertotti, 2003a, b, 2004; Signell et al., 2005; Bolaños-Sanchez et al., 2007; de León and Soares, 2008; de León et al., 2012), the Caribbean Sea and Gulf of Mexico (Appendini et al., 2013), and the Black Sea (Van Vledder and Akpınar, 2015) but not for the area of interest in the present study, i.e., that around the North and Baltic Seas. Hence, the accuracy of
the spectral wave model WAM is assessed for both normal and extreme conditions using different meteorological input data presently available. The sensitivity of the wave model to the temporal and spatial resolution of the meteorological input data is estimated.

Another way to increase the accuracy of the modelled significant wave height is by assimilating the significant wave height measured by satellites into a first-guess wave field (Thomas and Dwarakish, 2015). While altimeter data related to the open
ocean are of good quality and used routinely, for coastal areas, their quality tends to deteriorate, which results in systematic flagging of up to a few tens of kilometres from the coast (Cipollini et al., 2010; Vignudelli et al., 2011; Fenoglio-Marc et al., 2015). One issue in coastal altimetry is land contamination in the footprint of the altimeter due to different ocean and land surface reflectivities, leading to incorrect interpreted waveforms and therefore incorrect significant wave heights (Cipollini et al., 2010; Vignudelli et al., 2011). Hence, the advantage of improving the sea state by assimilating altimeter data into the
wave model cannot be employed close to a coast, where people are interested in accurate wave forecasting to protect and design



coastal structures such as, e.g., dykes (Thomas and Dwarakish, 2015). The difficulties in taking satellite measurements close to a coast, e.g., retracking at a land/sea interface, have already been reduced by CryoSat-2 and, even more so, by Sentinel-3A (Beneviste and Vignudelli, 2009). In this paper, the quality of the newly available Sentinel-3A data is analysed in comparison with the data from CryoSat-2 and Jason-2, especially that related to coastal areas. Then, the data are merged with the wave

model results to produce a best-guess wave field.

In the next section, the measured satellite and in situ data as well as the wind forcing data and the numerical wave model used are described (Sect. 2). This is followed by an assessment of the sensitivity of the wave model to different wind input data (Sect. 3). In Sect. 4, the quality of the newly available satellite data from Sentinel-3A with respect to that of older satellites is analysed. Then, the satellite and model data are combined to generate a best-guess wave field (Sect. 5). The summary and

conclusions are given in the last section (Sect. 6).

## 2   Data and Model

Here, the ocean wave model WAM is forced using different meteorological input data to evaluate the sensitivity of the model to different wind input spatial and temporal resolutions. Therefore, the numerical model and wind input data used are introduced in this section. Information regarding the in situ measurements used here is also given. Furthermore, the satellite data, especially

that of the new Sentinel-3A satellite, are presented.

### 2.1   Satellite altimeter data

In this study, wave height data derived from the Jason-2, CryoSat-2 and Sentinel-3A altimeter missions are used. Jason-2 is a classical pulse-limited altimeter operating in low-resolution mode (LRM) that was in operation, with a revisiting time of 10 days, from June 2008 to October 2016 (annonymous@avisoftp.cnes.fr).

The CryoSat-2 satellite, launched in April 2010, is the first space-borne instrument with synthetic aperture radar (SAR) capabilities. It can operate in one of three modes, i.e., SAR mode, interferometric SAR (SARIn) mode and low-rate mode (LRM), following a geographical mask, which is regularly updated. Compared to conventional pulse-limited (or conventional) altimetry (CA), SAR altimetry provides a better along-trajectory resolution and a higher signal-to-noise ratio (SNR). Over the northeastern Atlantic, CryoSat-2 operates in SAR mode. Data collected in SAR mode and processed similarly to LRM

mode data are called reduced SAR (RDSAR) data. We use CryoSat-2 RDSAR data (C2-RDSARRADS-1Hz) from the RADS (http://rads.tudelft.nl/rads/rads.shtm) database and SAR products from the grid processing on demand (GPOD) service at ESRIN (C2-SARGPOD-1Hz) (https://gpod.eo.esa.int).

Sentinel-3A, launched in February 2016, is the first satellite operating entirely in SAR mode. RDSAR products are also available. Essentially, the altimeter data are 1D profiles along the ground track of the satellite, with a footprint size of 1.5 km

up to 10 km depending on the sea state across the track. The resolution along the track of the satellite is approximately 7 km for 1 Hz measurements. Each track is repeated every 27 days, with a deviation of $\pm 1$ km in longitudinal direction. 'Ascending' passes are from south-southeast to north-northwest, whereas 'descending' passes are from north-northeast to south-southwest.

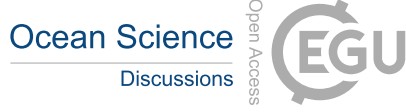

**Table 1.** Type and availability of the satellite data.

| Satellite | S | Mode | Period | Product Name |
|---|---|---|---|---|
| Jason-2 | J2 | LRM | 16.04.2016 - 25.11.2016 | J2-LRMAVISO-1Hz |
| CryoSat-2 | C2 | SAR | 01.01.2016 - 31.12.2016 | C2-SARGOPD-1Hz |
| CryoSat-2 | C2 | RDSAR | 31.12.2014 - 20.08.2017 | C2-RDSARRADS-1Hz |
| Sentinel-3A | S3A | RDSAR | 15.06.2016 - 15.11.2016 | S3A-RDSARNTC-1Hz |
| Sentinel-3A | S3A | SAR | 06.04.2016 - 20.08.2017 | S3A-SARRADS-1Hz |

In the present study the official Sentinel-3 SAR (S3A-SARNTC-1Hz) and RDSAR products (S3A-RDSARNTC-1Hz) are used, which are made available directly by Copernicus (https://sentinels.copernicus.eu/). The same data are available from RADS.

## 2.2 In situ measurements

In situ observations have great accuracy, but their geographical distribution is highly inhomogeneous, being mainly along coastal regions of industrialized countries. Gaps in measurements and other types of inhomogeneities also occur frequently in in situ observational records (Bidlot et al., 2002). While remote sensing measurements can be seen as a viable alternative to buoy observations, the shortness of the existing time series and the poor temporal resolution pose limitations to their use in wave climate studies (Stopa, 2018).

The results of the wave model and the satellite measurements are evaluated via a comparison with in situ observations at 165 locations. Most of the data are from the Global Telecommunication System (GTS), which were obtained by and are archived at the European Centre for Medium-Range Weather Forecasts (ECMWF) (Bidlot and Holt, 2006); other data were gathered by the ECMWF as part of the JCOMM Wave Forecast Verification project (Bidlot 2017). This data set was augmented with in situ wave buoy data provided by the Federal Maritime and Hydrographic Agency (Bundesamt für Seeschifffahrt und Hydrographie, BSH). Fig. 1 shows the locations of these in situ data. Moored wave data buoys are anchored at fixed locations and regularly collect observations from different atmospheric and oceanographic sensors. Moored buoys are usually deployed to serve national forecasting needs, to serve maritime safety needs or to observe regional climate patterns (http://www.jcommops.org/dbcp/platforms/types.html). Data are usually collected by either Argos, Iridium, ORBCOMM, GOES or METEOSAT, transmitted in real-time and shared on the GTS of the WMO. They are generally upgraded or serviced yearly. Over the North Sea and Norwegian Sea, the bulk of the data comes from the oil and gas industry, kindly supplied to the meteorological community via the GTS. Generally, the data are from instruments mounted on a platform or a rig. Note, however, that due to a lack of metadata in the GTS record, it is impossible to determine exactly which sensor was used. Wave height, wind speed and wind direction measurements are available every hour. Following a basic visual inspection of the data, the wave height measurements are collocated with the wave model simulations, using the grid point closest to the location of the in situ measurements. The wind measurements, however, have to be adjusted to a height of 10 m above the surface to compare the measurements with the model data. For the wind speed, the method used by Bidlot et al. (2002) is applied. With



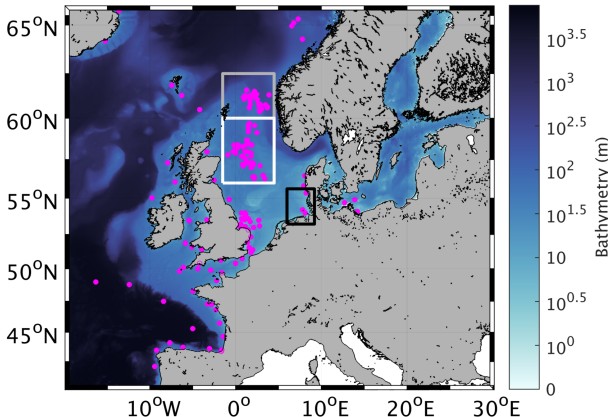

**Figure 1.** Bathymetry of the model area and locations of the GTS measurements.

the steady-state neutrally stable logarithmic vertical wind profile relation (Eq. 1), the friction velocity (u*) is calculated from the wind speed at the measurement height (U(z)) with the assumption that the surface roughness ($z_0$) can be specified by the Charnock relation (Eq. 2) with a constant parameter ($\alpha$) of 0.018 and g denoting the acceleration due to gravity. $\kappa$ in Eq. 1 is the von Kármán constant and has a value of 0.41. After obtaining u* via Eq. 1, the wind speed at z=10 m can be calculated
using the same equation.

$$U(z) = \frac{u*}{\kappa} ln \left( \frac{z}{z_0} \right) \tag{1}$$

$$z_0 = \alpha \frac{u*^2}{g} \tag{2}$$

### 2.3   Wave model WAM and meteorological input data used

The spectral wave model WAM Cycle4.6.2 is used here (WAMDI Group, 1988; Komen et al., 1994; Staneva et al., 2017).
The model runs as shallow waterversion taking into account depth refraction and wave breaking, and is therefore suitable for coastal applications. The 2D wave spectra are calculated on a polar grid with 30 directional 15° sectors and 30 frequencies logarithmically spaced from 0.042 to 0.66 Hz. A spherical grid is used for the spatial dimensions, with ∼0.06° resolution in the x-direction (east-west) and ∼0.03° resolution in the y-direction (north-south). The bathymetry and the study area are shown in Fig. 1. The forcing values at the open boundaries of the model domain are calculated via a coarser model simulation for the
whole North Atlantic driven by ERA-Interim winds.

To estimate the sensitivity of the wave model to the temporal and spatial resolutions of the meteorological input, different wind input data are used (Table 2). The ERA-Interim, ERA5 and coastDat-3 reanalyses, as well as the ECMWF operational





analysis/forecast and the German Weather Service (Deutscher Wetterdienst, DWD) forecast, are used as meteorological input data to force the wave model. ERA-Interim is a global reanalysis produced by the ECMWF (Dee et al., 2011). The temporal resolution of the output is six hours, and the spatial resolution is approximately 79 km (Berrisford et al., 2009). The downloaded data are interpolated to a spatial resolution of 0.125°. The successor of ERA-Interim is ERA5 (Hersbach and Dee, 2016). The spatial resolution of the model is increased to 31 km (0.28125°). The output is downloaded and interpolated to 0.25°. Furthermore, very important for the wave model simulations is that the temporal resolution of ERA5 is increased to an hourly one (ECMWF, 2017b). In addition, the ECMWF 6-hour operational analysis is used to force the wave model. When hourly temporal resolution is needed, the first twelve hours of the forecast wind fields from 0 and 12 UTC are taken, with the operational analysis at 0 and 12 UTC being used to initialize the forecast. The horizontal resolution is ∼9 km (ECMWF, 2017a) and is interpolated to a spatial resolution of 0.125° for the downloaded data. Aside from the wind input provided by the ECMWF, the hindcast coastDat-3 produced by the Helmholtz-Zentrum Geesthacht (HZG) using the COSMO-CLM model (Rockel et al., 2008; Geyer, 2014) is used to force the wave model. coastDat-3 has a temporal resolution of one hour and uses a rotated grid with a spatial resolution of 0.11° (HZG, 2017). Vertically, 40 levels up to an altitude of 22.7 km are used. As boundary conditions for the model domain, ERA-Interim is used. Another data set used to force the wave model is the DWD forecast, which is produced using the ICON_EU numerical model with a spatial resolution of 6.5 km and an output that is available every hour (Reinert et al., 2018). The impact of the temporal resolution of the wind forcing on wave simulations is evaluated in the next Section. Therefore, model experiments with six-hour wind forcing from ERA5 and the DWD forecast are conducted, with the wind data being updated every six hours based on the hourly output.

**Table 2.** Horizontal and temporal resolutions of the meteorological input data.

| Meteo Data Set | Resolution | |
| --- | --- | --- |
| | Horizontal | Time |
| ERA-Interim | 79 km x 79 km | 6 h |
| coastDat-3 | 0.11° x 0.11° | 1 h |
| ECMWF operational analysis/forecast | 9 km x 9 km | 6 h/1 h |
| ERA5 | 31 km x 31 km | 1 h/6 h |
| DWD forecast | 6.5 km x 6.5 km | 1 h/6 h |

## 3 Sensitivity of wave model to wind conditions

In this section, the sensitivity of the wave model to different spatial and temporal resolutions of the wind input data is analysed by assessing the general performance of the wave model under different wind forcings over the entire study period (from June to November 2016) and the entire model area. The quality of the simulated significant wave height during an extreme event is analysed in detail.



### 3.1 General performance of modelled waves and winds

#### 3.1.1 Significant wave height

To study the sensitivity of the wave model simulations to the wind conditions, WAM is forced using eight different wind data sets, as described in Sect. 2.3. The general performance of WAM on all different wind data sets is similar and good compared to the in situ observations (Fig. 2). Especially during calm conditions, the significant wave heights in the eight model experiments are similar. However, during extreme events, the differences in the simulated significant wave height become more apparent. Particularly, the WAM simulation with coastDat-3 wind forcing overestimates the large significant wave heights (Fig. 2b). Also, the simulation with hourly wind forcing of the DWD forecast tends to slightly overestimate the large significant wave heights (Fig. 2g). On the other hand, WAM forced using ERA-Interim, the ECMWF operational analysis/forecast and ERA5 wind data slightly underestimates the large significant wave heights with respect to the measurements taken at the GTS stations (Fig. 2a, 2c, 2d, 2e and 2f). Regarding the statistical values, the best wave model performance is seen in the simulation forced using the hourly ECMWF operational analysis/forecast atmospheric data. Using the DWD forecast as wind forcing data leads to a smaller bias. However, the root mean square error (RMSE) of 29.9 cm and the scatter index (SI) of 0.191 are the lowest, and the correlation coefficient (CORR) of 0.959 is the largest for the model simulations performed using hourly ECMWF operational analysis/forecast wind data. The differences in the statistical values for the results of WAM with the ECMWF operational anaylsis/forecast and ERA5 data are very small and approximately one magnitude less than the differences in the results produced with the ERA-Interim, coastDat-3 and DWD forecast wind forcings. Therefore, the model simulations with wind forcings of either the ECMWF operational analysis/forecast or ERA5 produce good results that are closer to the GTS measurements than the simulations with the other wind forcings. Notably, the model results corresponding to hourly wind input have better statistical values than the corresponding simulation with six-hour wind input (compare Fig. 2c to Fig. 2d. This once again justifies the crucial importance of using high-frequency wind forcing data (with a minimum of one hour) for wave simulations, especially for operational purposes.

#### 3.1.2 Wind input data

When comparing the wind speed with the in situ GTS measurements (Fig. 3), the best statistical values are achieved by ERA5 (Fig. 3d), although all performances are fairly similar. For this analysis, the original wind data are used; therefore, only the ERA-Interim data are taken every six hours, whereas all other wind data are taken every hour. For high wind speeds, a slight underestimation of the modelled wind speed compared to the GTS measurements still occurs. However, this underestimation reflects a large improvement compared to the underestimation found by Cavaleri and Bertotti (2003b). The overprediction of coastDat-3, which can be seen for high significant wave heights, is not evident for the magnitude of the wind in the wind forcing (Fig. 3b). One possible reason for the higher significant wave heights during extreme events might be the wind direction, which has a bias of approximately 12° for the coastDat-3 data. Hence, the wind direction is shifted to the right, affecting the fetch length in the North Sea, especially for northwesterly wind directions. For the other wind data, the bias of the wind direction



is only approximately 1° to 2°. Since the fetch in coastal areas is limited because of the presence of land, this shift in wind direction can impact the simulated significant wave height.

The general performances of WAM under all different wind forcings are good and fairly similar, especially under calm conditions, where no major differences are found. During extreme events, however, the model simulations tend to be spread

out, with the coastDat-3 wind forcing overestimating and the ERA-Interim, ECMWF operational analysis/forecast and ERA5 wind forcings underestimating the large significant wave heights. In the wind data, this cannot be found. The wind is only very slightly underestimated. Particularly, the overestimation of the significant wave height with the coastDat-3 wind forcing cannot be found in the wind data.

### 3.2  Evaluation of the ensemble during an extreme event

As described in the previous section, the modelled significant wave heights tend to spread out during extreme events for different model experiments. Here, a more detailed analysis of data variability during an extreme event is provided. During the study period from June to November 2016, an extreme event occurred on 29th September 2016. The centre of the low pressure system was located along the coast of Norway. Thus, the highest wind speeds occurred in the northern part of the North Sea, and the corresponding highest significant wave heights could be found in the northern part of the North Sea. At 11 UTC, the

area with maximum significant wave height coincided with the locations of the GTS measurements. Hence, this time step is chosen for further analyses.

### 3.2.1  Significant wave height of each ensemble member

In Fig. 4, the wave height estimates of each ensemble member for 29 September 2016, 11 UTC, are shown together with the locations of the GTS measurements. The horizontal patterns of the eight model runs for this extreme event are quite different.

The largest significant wave height is found in the model simulation with the coastDat-3 wind forcing of more than 9 m (Fig. 4b). The smallest maximum significant wave height is found for the model simulation with the six-hour ECMWF operational analysis wind forcing (Fig. 4d). Notably, the maximum of the model simulation with six-hour wind forcing (Fig. 4a, 4d, 4f and 4h) is shifted further to the east than in the model simulations with hourly wind input (Fig. 4b, 4c, 4e and 4g). Furthermore, in the model simulations with the six-hour wind input, the maximum of the significant wave height is smaller than that with the

hourly wind input. This again emphasizes the importance of using higher-time-frequency wind data for wave simulations over the study area.

When comparing the modelled significant wave height with the GTS measurements in the northern part of the North Sea (55° N 2° W to 62.5° N 5° E, white and gray box in Fig. 1), none of the simulations is perfectly in line with the measurements, but the model simulation with the hourly ERA5 wind forcing has a bias of only -0.02 m and an SI of 0.144 (Fig. 4e). For the

model simulation with the ERA5 wind forcing, the RMSE is 0.56 m, which is smallest compared to that of the other model experiments. The only simulation with a smaller SI is the run with the hourly ECMWF operational analysis/forecast wind forcing, achieving a value of 0.139 (Fig. 4c). The bias, though, is 0.1843 m, which is clearly larger than the bias for the model simulation with the hourly ERA5 wind forcing. The model experiment simulation with the six-hour ERA5 wind forcing has



the largest SI (0.193) as well as the largest RMSE (Fig. 4f). Compared to the GTS measurements, the simulations with the ERA-Interim, six-hour ECMWF operational analysis and both ERA5 wind forcings underestimate the significant wave height, with the largest underestimation (0.57 m) being made by the model simulation with the six-hour ERA5 wind forcing (Fig. 4f). The model simulations with the coastDat-3, hourly ECMWF operational analysis/forecast and both DWD forecasts all

overestimate the significant wave height in the northern part of the North Sea by up to 1.28 m for the case of coastDat-3 wind forcing (Fig. 4b).

### 3.2.2   Empirical orthogonal functions

To study the variance of the significant wave height of the eight ensemble members during the extreme event, an empirical orthogonal function (EOF) analysis of the extreme event on 29th September 2016, 11:00 UTC, is performed. The EOF analysis

is carried out as described by Björnsson and Venegas (1997).

Fig. 5a shows the mean of the ensemble depicted in Fig. 4. The associated standard deviation with respect to the mean is given in Fig. 5b. Clearly, the largest difference between the ensemble members is located in the northern part of the North Sea. The ensemble members also differ substantially with respect to the local wave height maximum off the coast of Iceland.

The first EOF of the significant wave height represents 56.16 % of the total variance of the ensemble. The maximum variance

is found in the area of the maximum significant wave height in the northern part of the North Sea (Fig. 5a and 5c). This demonstrates that the largest difference between the different model simulations is the magnitude of the significant wave height peak. In this case, the model simulation with the coastDat-3 wind forcing has the highest simulated significant wave height maximum (9.5 m), and the model simulation with the six-hour ECMWF operational analysis wind forcing, the lowest simulated significant wave height maximum (6.6 m).

The maximum of the second EOF of the significant wave height, which represents 19.31 % of the total variance, is located in the northern part of the model domain near the coast of Iceland (Fig. 5d), which overlaps the area of the second maximum significant wave height (Fig. 5a). This shows that the model simulations also differ in terms of the magnitude of the maximum significant wave height in the northern part of the model domain. In this area, the significant wave height in the model simulations with the ERA-Interim and coastDat-3 forcings is clearly larger than that in the model simulations with the other wind

forcings. These two differences are also found regarding the standard deviation of the ensemble. Combining the first two EOFs explains 75.47 % of the total variance of the ensemble. However, with the EOF, more differences in the model simulations can be found.

The third EOF pattern shows a dipole in the northern part of the North Sea (Fig. 5e). This means that in the model simulations, the exact positions of the significant wave height maximum differ. The orientation of the dipole is in the east-west direction and

therefore represents the variation of the peak location in the different model simulations in the zonal direction. In this context, larger differences are especially found between the model simulations with the hourly and six-hour wind forcings, with a peak shift of approximately 290 km. The third EOF represents 9.98 % of the total variance.

The fourth EOF explains 7.71 % of the total variance. This EOF reveals the larger-scale differences in the synoptic situation and, therefore, in the wind fields, which are also reflected in the wave field. In the wind forcing data, the exact location of the





centre of the low-pressure system and, therefore, the area of light wind differs, which also leads to different wave heights off the coast of the northern part of Norway. In addition, due to the different strengths of the wind fields in the wind forcings, the significant wave height west of Ireland in the Atlantic as well as off the coast of Norway is larger relative to that east of Great Britain due to the fetch conditions (Fig. 5f).

### 3.2.3   Time series of significant wave height, wind speed and wind direction

Since the largest difference in significant wave height between the ensemble members is the magnitude of the respective peak, further investigation is required. Time series extracted from the ensemble members are compared to the time series of the GTS measurements (Fig. 6). For this analysis, the mean of the GTS measurements in the northern part of the North Sea (55° N 2° W to 60° N 5° E, white box in Fig. 1) at each time step is taken, and the standard deviation is calculated to estimate the variation of the measurements within the considered area. The same is done for the significant wave height of each ensemble member at the locations of the GTS measurements. Here, only the southern part of the in situ measurements in the northern part of the North Sea is taken, as the time series of the northern and southern parts differ due to the centre of the low-pressure system passing over only the northern part of the in situ measurements. Therefore, the mean is taken for in situ measurements with similar temporal behaviours.

Fig. 6a depicts the spread of the simulated significant wave heights between the experiments with different wind forcings. During the extreme event, the maximum significant wave height varies between 4.7 m for the simulation with the six-hour ERA5 wind forcing and 6.9 m for the simulation with the coastDat-3 wind forcing. The observed significant wave height from the GTS measurements lies in between the two extremes at 5.3 m. Therefore, coastDat-3 overestimates the significant wave height by approximately 1.6 m. During this extreme event, the overestimation is mainly due to coastDat-3 overestimating the wind speed at that time (Fig. 6b). Also, in coastDat-3, the wind direction is shifted in the clockwise direction by approximately 12° for the majority of time during this extreme event (Fig. 6c). This impact is likely to be small compared to the overestimation of the wind speed, as the fetch is rather limited with respect to the wind directions between south and west-northwest. The wind direction in other areas may affect the significant wave height in this area, though, due to swells travelling into the analysed area. For coastDat-3, the area affected by high wind speeds and, therefore, also by high significant wave heights is larger than that for the other wind forcings (Fig. 4b). This might also contribute to the high significant wave height shown in Fig. 6a, as the values averaged in this analysis cover the northern part of the North Sea.

The model experiment with the six-hour ERA5 wind forcing yields the lowest significant wave heights for 29th September 2016 (Fig. 6a). In this simulation, the peak is underestimated by approximately 0.6 m. This underestimation of the significant wave height is also due to the underestimation of the wind speed (Fig. 6b). Since WAM receives the wind data only every six hours, the wind speed peak is missed in the wind forcing; therefore, the peak in terms of the significant wave height is omitted. This problem can also be seen for the model simulations with the ERA-Interim and six-hour ECMWF operational analysis wind forcing. Although the wind speed of the hourly DWD forecast and ECMWF operational analysis/forecast matches the observed wind speed very well (Fig. 6b), WAM overestimates the peak in the significant wave height (Fig. 6a). This might indicate that WAM needs to be further tuned regarding the significant wave height during extreme events. Another possible



reason for this overestimation could be the swells travelling into the area.To clearly conclude either reason, more extreme events need to be analysed. For this extreme event, WAM simulates the maximum significant wave height two hours earlier, even though the timing of the wind speed peak fits well for the two wind forcings.

The peak in the observed significant wave height is best illustrated by the model simulation with the hourly ERA5 wind
forcing (Fig. 6a). The maximum significant wave height differs by only approximately 0.01 m. However, in this run, similar to the simulations with the coastDat-3 data, hourly DWD forecast and hourly ECMWF operational analysis, the simulated peak in the significant wave height occurs two hours earlier than the observed peak. The model simulation with the six-hour DWD forecast wind forcing slightly overestimates the observed peak (Fig. 6a), although the maximum wind speed is below the maximum observed wind speed (Fig. 6b). The period of the peak for all model simulations with the six-hour wind forcing in
terms of the significant wave height is longer than that for the model simulations with hourly wind forcing. Here,period of the peak is estimated based on the time at which the significant wave height exceeds 99 % of the peak value. For this significant wave height peak,the period of the peak for the model simulations with hourly wind forcing is one hour, whereas for the model simulations with six-hour wind forcing, the period of the peak is three to four hours.

One and two days earlier, two smaller wave height peaks occur. The first one on 27th September 2016 is overestimated by all
of the model experiments, although the corresponding peak in the wind speed is captured well by the model simulations with the hourly ERA5 and ECMWF operational analysis/forecast wind forcings. The six-hour wind forcings capture this peak very well, but due to the wind speed being high three hours prior to and after the peak, the simulated significant wave height is too high. The model simulation with the hourly DWD forecast wind forcing is the most successful at reproducing the significant wave height peak, although the estimated wind speed is lower than the observed wind speed. The second peak is best matched
by both model simulations with the ECMWF operational analysis/forecast wind forcing. Both simulations with ERA5 wind forcings slightly underestimate the significant wave height peak. All other simulations overestimate the significant wave height.

During calm conditions both before and after the peaks, the results of all model simulations are very similar.

From the analyses above, it can be concluded that during extreme events, the wave model results are quite sensitive to the wind forcing. Hence, high-quality wind data are needed to improve the ability to predict the sea state. For our area of
interest, a higher temporal resolution of the wind forcing is more important than a higher spatial resolution. Although the spatial resolution of the DWD forecast and coastDat-3 is higher than that for ERA5 and the ECMWF operational analysis, the wave model simulations using the latter two increase the model capabilities. However, clearly better results can be found via model simulations with hourly wind forcing than via those with six-hour wind forcing. This conclusion differs from that of the study on the Black Sea by Van Vledder and Akpınar (2015). Notably, the different spatial resolutions tested are produced by
different atmospheric models or model setups, which can also lead to differences. Therefore, the differences cannot only be traced back to the different spatial resolutions. In our study, the hourly ECMWF operational analysis/forecast as well as the hourly ERA5 wind forcing produce the best results, as they are more similar to the observations made during the extreme event at the end of September 2016. Also, the statistical values for the entire study period and over the study area are better for the model simulations forced with hourly ERA5 and ECMWF operational analysis/forecast than for the model simulations with





the six other wind data sets. Under calm conditions, the model simulations with all eight wind forcings produce fairly similar results.

## 4    Comparison of Satellite Data

In this section, the quality of the newly available Sentinel-3A satellite data is assessed and compared to that of older satellite

data. The focus in this study is on coastal areas, where the quality of both the satellite as well as the model data tends to deteriorate. Also, the quality of the Sentinel-3A data is analysed based on the relative orientation of the coastline and satellite heading, varying metocean conditions, and the wind direction relative to the satellite flight direction. In this section, when comparing satellite data with the simulated significant wave height, the model simulation with the ERA5 wind forcing is used, as this simulation, along with that with the ECMWF operational analysis/forecast wind forcing, produced the best results

during both extreme events and calm conditions (Section 3).

### 4.1    General quality of measured significant wave height

To estimate the overall performances of the different satellite products during the entire study period and over the study area, scatter plots of the in situ measurements versus remote sensing measurements are analysed (Fig. 7). For these comparisons, the satellite data are allowed to have a maximum spatial distance of 20 km and a maximum time gap of 30 min with respect

to the in situ measurements (Fenoglio-Marc et al., 2015). The general performances of all five satellite products are good and very similar. The correlation between all products varies by only 3 % with values ranging from 94 % to 97 %. The SI is the largest for the CryoSat-2 RDSAR product, being approximately 0.22. For the SAR products of Sentinel-3A and CryoSat-2 as well as for Jason-2, the SI is approximately 0.17. However, the satellites tend to overestimate the significant wave height of in situ measurements, especially Sentinel-3A SAR and both CryoSat-2 products, with biases of up to 26 cm. The smallest bias

(only 6 cm) is found for the Jason-2 measurements.

### 4.2    Scatter index along the satellite track

To analyse the spatial distribution of the quality of the satellite data, the SI between the modelled and measured significant wave heights along the satellite tracks within each grid box is calculated for Jason-2 and Sentinel-3A SAR (Fig. 8). Since very few data exist within each grid box during the study period, for this analysis, the study period is extended to the end of August

2017 to achieve a more robust SI result. For both satellites, the SI is small over the open ocean and becomes larger closer to the coast. Notably, in coastal areas, the SI for Sentinel-3A SAR is smaller than that for Jason-2. Especially in the northern part of the Baltic Sea, the Danish Straits and along the coastal areas of the southern North Sea, the SI is reduced for Sentinel-3A SAR compared to that for Jason-2. This clearly indicates that Sentinel-3A SAR performs better over coastal areas than Jason-2.

To quantify this, the statistical values within the first 10 km off the coast are calculated for all three different satellites (Table

3). In some earlier studies, this area was neglected due to the deteriorating quality of the satellite data (Fenoglio-Marc et al., 2015). For Sentinel-3A, the RMSE is reduced by approximately 0.1 m and the SI is reduced by 0.17 compared to the values for



the other two satellites. The bias is reduced by 0.08 m compared to that for Jason-2 and 0.16 m compared to that for CryoSat-2.
The correlation for Sentinel-3A is increased by 10 % compared to that for Jason-2 and 5 % compared to that for CryoSat-2.
Furthermore, the statistics of Sentinel-3A within the first 10 km are closer to those over the whole study area, which is not the
case for the the other two satellites (Table 3; see Fig. 7). This indicates that the quality of the data of Sentinel-3A over coastal

areas is closer to that over the open ocean compared to the data quality of CryoSat-2 and Jason-2.

**Table 3.** Comparison of the data quality within the first 10 km off the coast for all three satellites.

|  | RMSE (m) | SI | Bias (m) | Correlation |
|---|---|---|---|---|
| Jason-2 | 0.5219 | 0.4977 | 0.2461 | 0.8075 |
| CryoSat-2 SAR | 0.4860 | 0.4957 | 0.3334 | 0.8548 |
| Sentinel-3A SAR | 0.3985 | 0.3324 | 0.1682 | 0.9138 |

## 4.3   Comparison of data quality for onshore and offshore flights

Due to the way satellite altimeter data are processed, the data quality can deteriorate in the vicinity of coastlines, particularly
for passes from land to ocean. To test how much the satellite measurements over the study area are affected by this problem,
the flights are separated into onshore and offshore flights, with onshore flights passing from the ocean to the shore and offshore

flights passing from the shore to the ocean. For the analysis here, again, only measurements within the first 10 km off the coast
are taken. When comparing the statistical values for Sentinel-3A SAR for both onshore and offshore flights, no substantial
differences are found, and the statistical values are very similar (Tab. 4). Therefore, the transition from land to water does not
influence the quality of the satellite observations over our study area.

**Table 4.** Comparison of the data quality, organized by onshore and offshore flights, for Sentinel-3A SAR. Only measurements taken within
the first 10 km off the coast are used.

|  | RMSE (m) | SI | Bias (m) | Correlation |
|---|---|---|---|---|
| onshore | 0.3877 | 0.3244 | 0.1666 | 0.9151 |
| offshore | 0.3981 | 0.3219 | 0.1695 | 0.9195 |

## 4.4   Comparison of data quality for long- and short-fetch conditions

Another assessment of the quality of the data measured by the satellites can be carried out by analysing their quality in terms
of the fetch conditions. To test this, Sentinel-3A SAR data within the German Bight ($53.23°$ N $6°$ E to $55.62°$ N $9.2°$ E, black
box in Fig. 1) are split into two groups: that for long-fetch situations and that for short-fetch situations. Long-fetch situations
within the German Bight are characterized by northwesterly winds, while short-fetch conditions occur for southeasterly winds.



The analyses demonstrate that the data quality for both situations is very similar (Tab. 5). The SI and the correlation have better values for long-fetch situations. The correlation between modelled and measured significant wave heights for long-fetch situations is 98 %, being 4 % larger than that for short-fetch situations. The SI for long-fetch situations is 0.09. The SI for short-fetch situations is approximately twice as large, i.e., 0.19. The RMSE and the bias, though, are better under short-fetch conditions. The RMSE, which is 21.6 cm for short-fetch situations, is approximately 16 cm smaller under short-fetch conditions than under long-fetch conditions. The bias under short-fetch conditions is only 0.7 cm. This is due to the over- and underestimation of the measured data essentially cancelling each other. Under long-fetch conditions, this is not the case, as the bias amounts to 33 cm. When analysing all directions, the statistical values lie between those under long- and short-fetch conditions. Hence, it can be concluded that the satellite measurements do not yield clearly better results for any of the conditions.

**Table 5.** Comparison of the data quality, organized by long- and short-fetch situations within the German Bight, for Sentinel-3A SAR.

|                  | RMSE (m) | SI     | Bias (m) | Correlation |
|------------------|----------|--------|----------|-------------|
| long fetch (NW)  | 0.3796   | 0.0943 | 0.3299   | 0.9809      |
| short fetch (SW) | 0.2164   | 0.1854 | 0.0065   | 0.9411      |
| all directions   | 0.2763   | 0.1658 | 0.1660   | 0.9524      |

### 4.5 Comparison of data quality for different relative wind and flight directions

In previous studies, e.g., Chelton and Freilich (2005), a dependency of the data quality on the wind and wave direction relative to the movement of a satellite was found, as satellites move while measuring the wind and wave conditions. Therefore, in this analysis, the measured significant wave height data are separated in terms of the wind direction relative to the satellite track. A slightly smaller RMSE, SI and bias can be found in situations where the wind comes from the direction opposite that of satellite motion. The best correlation, though, is achieved under cross-wind conditions, having a value of 96.7 %. Since the differences between all situations are quite small, i.e., 1.3 % for the correlation, 6 cm for the bias, 0.009 for the SI and 6.6 cm for the RMSE, the difference in the statistical values for all three situations cannot be regarded as substantial. Therefore, it can be concluded that the quality of the Sentinel-3A measurements does not depend on the wind direction relative to the satellite flight direction.

The newly available Sentinel-3A data yield better results for coastal areas compared to the data quality of older satellites such as Jason-2 and CryoSat-2. For Sentinel-3A, no substantial differences are found regardless whether the satellites pass from land to water or vice versa. Furthermore, the quality of the Sentinel-3A data does not differ substantially under either long- or short-fetch conditions within the German Bight. When comparing the data quality based on the wind direction relative to the satellite flight direction, again, no substantial differences can be found.



**Table 6.** Comparison of the data quality, organized by the wind direction relative to the satellite flight direction, for Sentinel-3A SAR.

|                | RMSE (m) | SI     | Bias (m) | Correlation |
|----------------|----------|--------|----------|-------------|
| along-wind     | 0.4396   | 0.1643 | 0.2794   | 0.9645      |
| opposing-wind  | 0.3757   | 0.1553 | 0.2254   | 0.9544      |
| cross-wind     | 0.4416   | 0.1625 | 0.2886   | 0.9673      |

## 5   Synergy of Satellite Data and Model Ensemble

To enhance the quality of the significant wave height data of the ensemble mean, the satellite measurements and the ensemble of the modelled significant wave height are combined to produce a best-guess wave field using the EOFs. A more detailed explanation of this method, which is based on a maximum a posteriori approach, can be found in Schulz-Stellenfleth and

Stanev (2010). The technique is illustrated for the extreme event on 29th September 2016, 11 UTC (Fig. 9). When comparing the ensemble mean of the significant wave height (Fig. 5a) to the GTS measurements in the northern part of the North Sea (55° N 2° W to 62.5° N 5° E, white and gray box in Fig. 1), where the maximum in significant wave height occurs, both are found to be in very good agreement, with an SI of 0.139, a bias of 0.11 m and an RMSE of 0.56 m. When using the satellite data with the satellite measurement standard deviation as an observation error, and when no bias correction is performed, the

statistical values of the best-guess wave field in terms of the GTS measurements become worse compared to the ensemble mean. To force the analysis to stay close to the already good ensemble mean, a rather high value of 3 m is assumed for the observation errors. The significant wave height reconstructed using the EOFs and the satellite measurements then have an SI of 0.138, a bias of 0.36 m and an RMSE of 0.65 m with respect to the GTS data (not used for the reconstructed significant wave height). As this is still not superior to the ensemble mean, a bias correction of the satellite measurements is carried out. The

reconstructed significant wave height (Fig. 9) then has the same SI as that before the bias correction, but the standard deviation of the error is reduced from 0.70 m to 0.65 m, and the bias and RMSE are improved to 0.06 m and 0.54 m, respectively. For this extreme event, the results demonstrate that a bias correction is absolutely necessary before assimilating the satellite data into a wave model. The analyses show that the model can be guided towards the right direction by the satellite data but that the satellite data are still not accurate enough compared to the in situ observations to be used to strictly force the model towards

the satellite observations.

## 6   Summary and Conclusions

In this study, the sensitivity of the wave model to wind forcing data with different spatial and temporal resolutions is tested. The analysis shows that the general performances of WAM for all different wind forcings are good and fairly similar. Especially during calm conditions, no major differences can be found. During extreme events, however, the model simulations tend to

be spread out, with the model simulation with the coastDat-3 and DWD wind forcings tending to overestimate the significant





wave height and the model simulations with the ECMWF operational analysis/forecast, ERA-Interim and ERA5 wind forcings tending to underestimate the high significant wave heights. The EOF analysis shows that the largest differences between the model simulations is the magnitude of the peak significant wave height, with a difference of 2.92 m between the smallest and largest significant wave height peaks. Also, the location of the maximum differs, especially between the simulations with hourly

and six-hour wind forcings, with approximately 290 km between the peaks. Furthermore, the larger-scale wind conditions change the wave conditions. The analysis of the time series clearly shows that hourly wind forcing data are needed to simulate the significant wave height peak correctly, as a six-hour wind forcing often misses the wind speed peak and, therefore, also the significant wave height peak. The best results are obtained by the simulation with the ECMWF operational analysis/forecast and ERA5 wind forcing. Further improvements in wave forecasting may be possible using a coupled wave-atmosphere model,

depending on the atmospheric model used.

Furthermore, the quality of the newly available Sentinel-3A data is assessed in comparison with data from older satellites, i.e., Jason-2 and CryoSat-2. The general performances are good and fairly similar between all satellite products, although all products tend to overestimate the in situ significant wave height measured within the GTS. The analysis of the spatial distributions of the satellite data quality reveals better results for Sentinel-3A over coastal areas than for Jason-2. Especially

within the first 10 km off the coast, these differences become apparent. In further analyses, no substantial differences between onshore and offshore satellite flights as well as for different metocean conditions can be found. Also, the satellite data quality does not depend on the wind direction relative to the flight direction. Therefore, Sentinel-3A has a clear advantage over the other satellites when utilized over coasts, exhibiting better skills than those of the other satellites compared to the wave model.

In the last section, where the ensemble and satellite data are merged, the carrying out of bias correction before assimilating

satellite data into a wave model is shown to be necessary. Also, for an extreme event, satellite data can be used to guide an ensemble towards a better best-guess wave field, though it cannot be used to strictly force the ensemble towards the satellite data, as they are not accurate enough compared to the in situ measurements.

*Competing interests.* The authors declare that they have no conflict of interest.

*Acknowledgements.* This publication has received funding from the European Union's H2020 Programme for Research, Technological

Development and Demonstration under grant agreement no. H2020-EO-2016-730030-CEASELESS. Luciana Fenoglio acknowledge the support of the European Space Agency (ESA) within the project SAR Altimetry Coastal & Open Ocean Performance (SCOOP). The authors would like to thank Beate Geyer for providing coastDat-3 wind data.

(c) Author(s) 2018. CC BY 4.0 License.



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




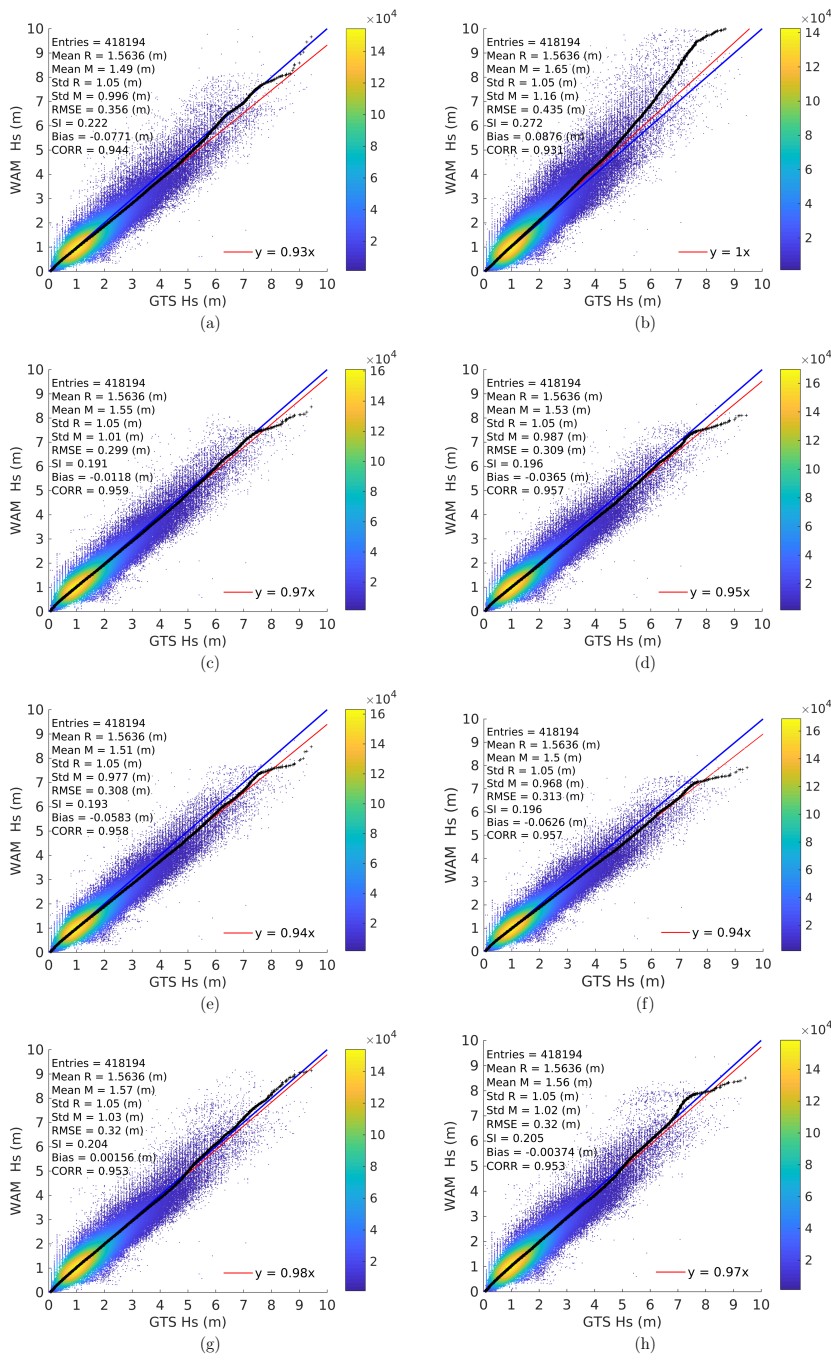

**Figure 2.** QQ scatter plot for measured (GTS wave data) and computed (WAM) significant wave heights with (a) ERA-Interim, (b) coastDat-3, (c) hourly and (d) six-hour ECMWF operational analysis/forecast; (e) hourly and (f) six-hour ERA5; and (g) hourly and (h) six-hour DWD forecast wind forcings from June to November 2016: QQ plot (black crosses), 45° reference line (blue line) and least-squares best-fit line (red line).





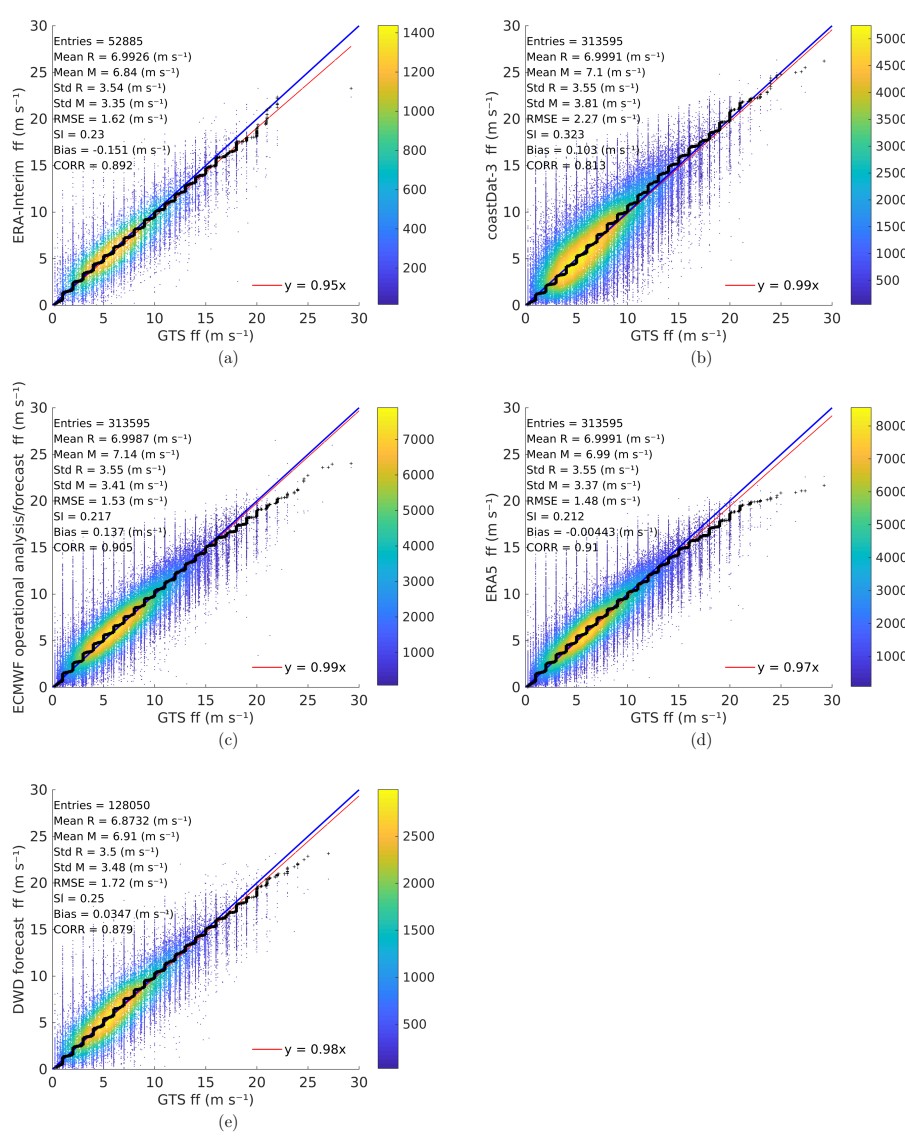

**Figure 3.** QQ scatter plot for measured (GTS wave buoys) and computed wind speeds from (a) ERA-Interim, (b) coastDat-3, (c) ECMWF operational analysis/forecast, (d) ERA5 and (e) DWD forecast from June to November 2016: QQ plot (black crosses), 45° reference line (blue line) and least-squares best-fit line (red line).





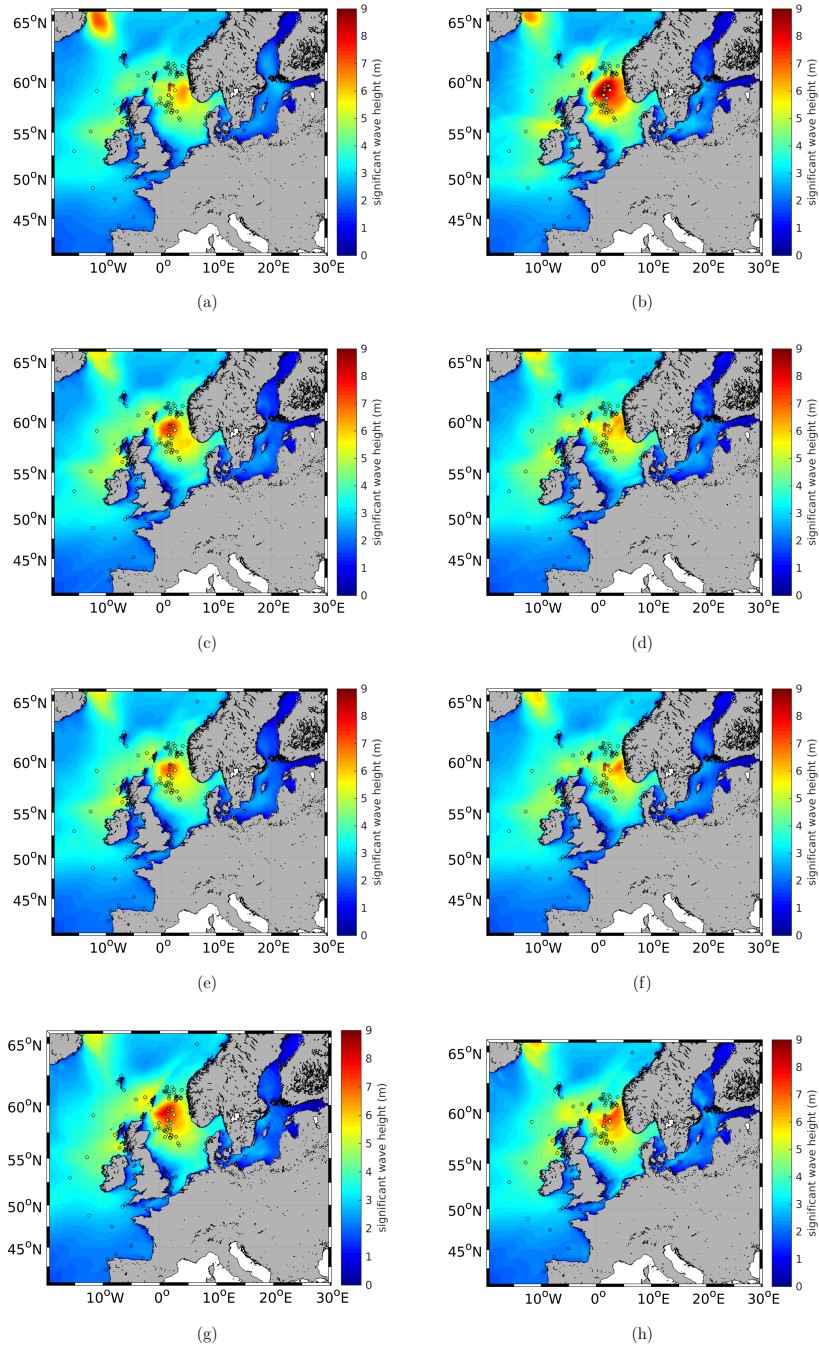

**Figure 4.** The significant wave height (m) of the ensemble for 29 September 2016, 11 UTC, as well as the GTS measurements for the model simulations with the (a) ERA-Interim, (b) coastDat-3, (c) hourly and (d) six-hour ECMWF operational analysis/forecast; (e) hourly and (f) six-hour ERA5; and (g) hourly and (h) six-hour DWD forecast wind forcings.




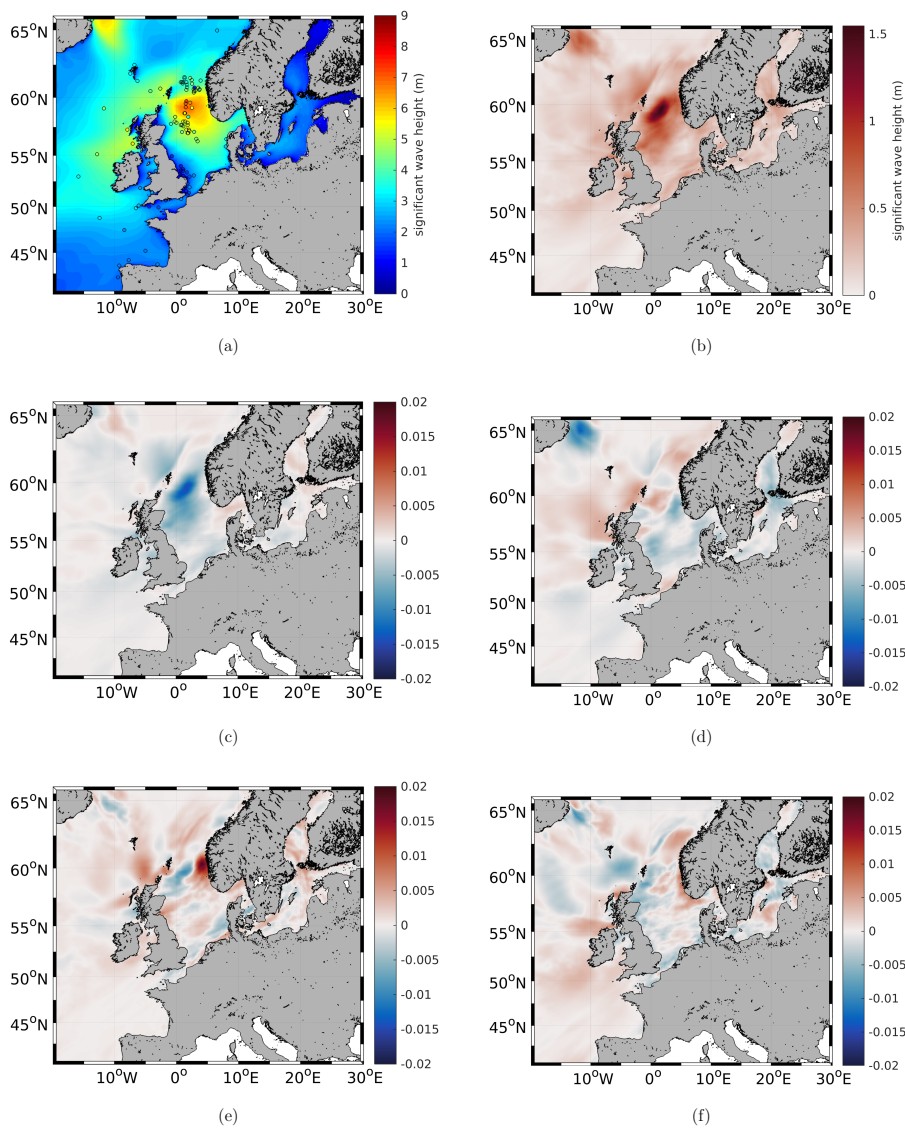

**Figure 5.** (a) The mean significant wave height (m) of the ensemble for 29 September 2016, 11 UTC, as well as (b) the standard deviation and the EOFs representing (c) 56.16 %, (d) 19.31 %, (e) 9.98 % and (f) 7.71 % of the total variance.





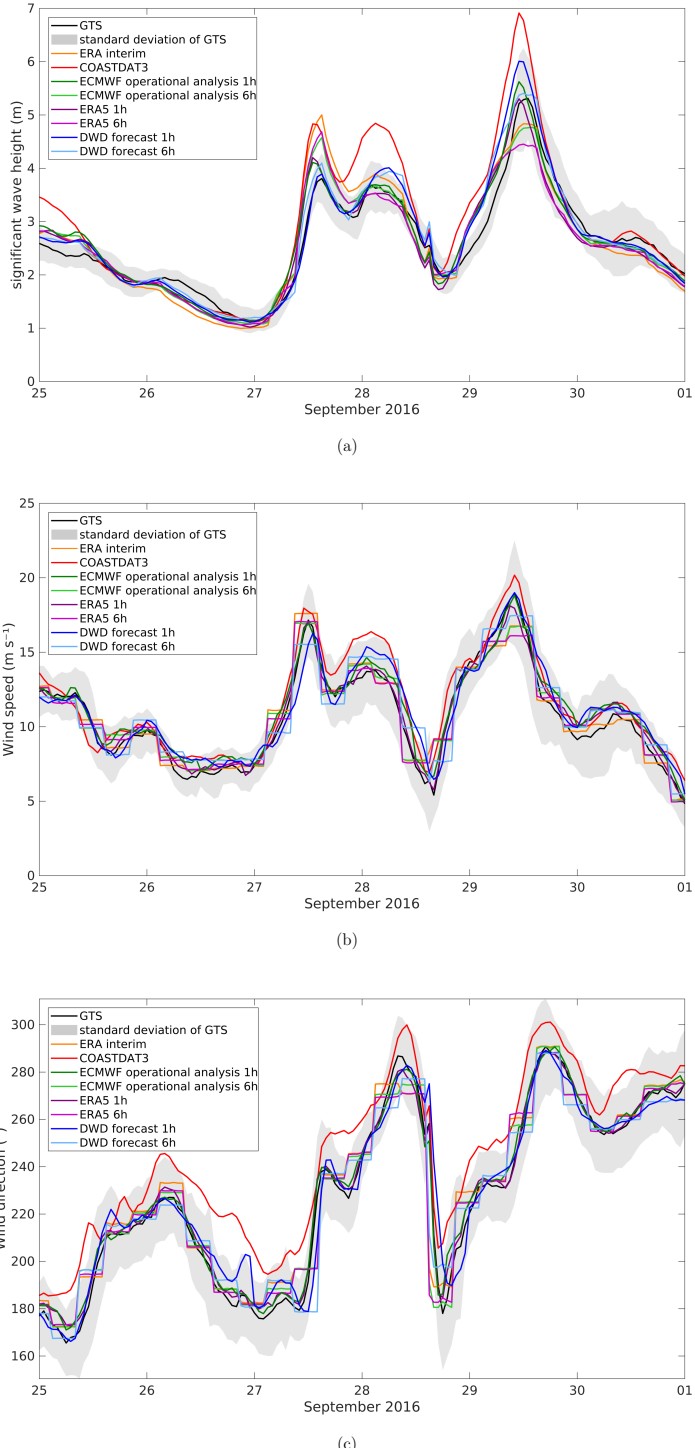

**Figure 6.** Time series of significant wave height (m) as modelled by WAM with different wind forcings and GTS measurements within the northern part of the North Sea (55° N 2° W to 60° N 5° E, white box in Fig. 1).





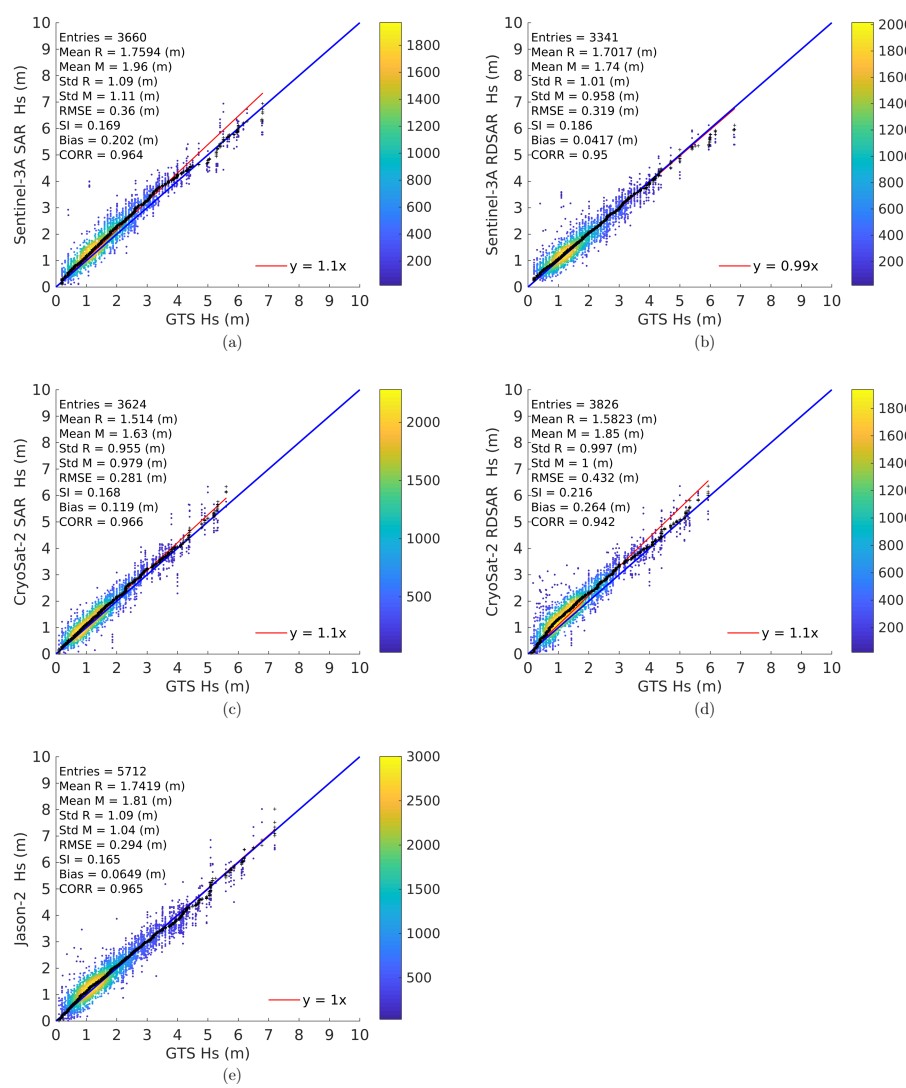

**Figure 7.** QQ scatter plots of measured significant wave height – in situ GTS vs. remote sensing data of (a) Sentinel-3A SAR, (b) Sentinel-3A RDSAR, (c) CryoSat-2 SAR, (d) CryoSat-2 RDSAR and (e) Jason-2 from June to November 2016: QQ plot (black crosses), 45° reference line (blue line) and least-squares best-fit line (red line).



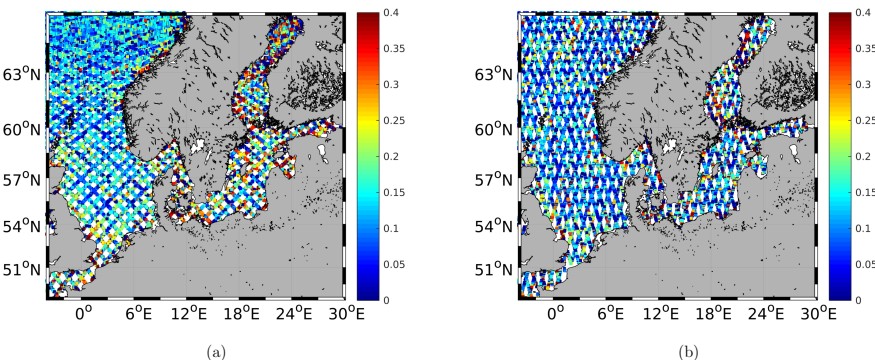

(a)  (b)

**Figure 8.** Scatter index between satellite and modelled significant wave heights along the satellite tracks for (a) Jason-2 and (b) Sentinel-3A SAR.

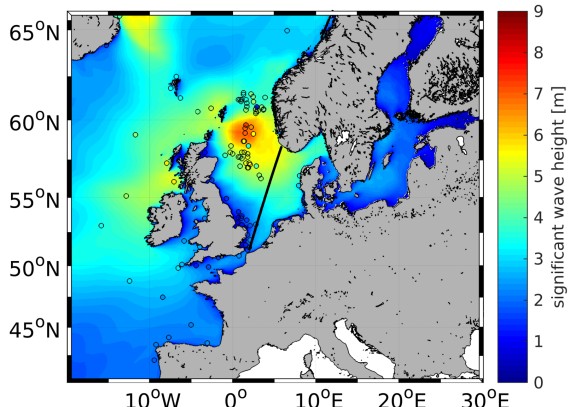

**Figure 9.** Best guess of the significant wave height of the ensemble, together with the Sentinel-3A data, on 29th September 2016 at 11 UTC.