# Peer review of "Synergy of wind wave model simulations and satellite observations during extreme events"

_Ocean Science, 2018_

## Referee Comment (RC1) · Anonymous Referee #1 · 3 Sep 2018

Overview:

This paper focusses on the issue of improving wave model performance by improving the driving wind fields. It is following many other papers on this topic but does add some originality in its assessment of the differences between wave model outputs. The wave model outputs are compared with in situ data as the ground truth. The results are similar to previous work in demonstrating that wave model results are sensitive to the wind forcing, which is not surprising or very new. A secondary part of the work is in examining the quality of satellite data on significant wave height, showing that the most recent satellite Sentinel-3A is superior to previous satellites, especially within 10km of the coast, but still requires bias correction of the significant wave height before any improvement in wave model performance could be achieved by its assimilation into an

operational model.

General:

The paper is generally clearly written and readable, apart from a few errors (listed below). However it could be improved by some reorganisation and rationalisation. The motivation for the study and the order of the presentation could be sharpened up. The satellite data assessment is stated to be the main motivation but the bulk of the paper addresses the wave model forcing by different atmospheric model wind fields and assessing which performs better. This exercise is somewhat flawed as the wind fields come from different models and it is not clear what are the differences between these. This should be discussed further. Some are reanalysis products and some are forecasts. Presumably these have different wind datasets assimilated into them. Thus it is not really a like-for-like comparison. The issue of model spatial and temporal resolution could be more rigorously assessed by sampling the same wind field at different resolutions, but this is not done. The authors state that the WAM model performs well with all the datasets, which has already been demonstrated. It is also known that higher spatial and temporal resolution improves the wave model performance. There is an interesting comparison between results, using an EOF analysis to show differences in wind speed, direction and location of the maximum affecting the maximum wave height in an extreme event. Using reference to a model ensemble where it is demonstrated that some wind fields are better quality than others seem somewhat perverse – can this be justified? Overall this paper needs moderate revision before acceptance for publication.

Detailed comments:

1. The title should include 'of wind and waves' after 'simulations'

2. The first line of the abstract is somewhat misleading as only satellite wave data are evaluated. Also this part of the work is second to the study of the spatial and temporal resolution of the wind forcing for the wave model and so perhaps should not

be introduced first.

3. P 2 line 7, suggest inserting 'in determining' to replace 'of'

4. P 2 line 31 'flagging of (data)' – explain further, this is rather cryptic.

5. P 5 line 11 – 30 times 15 does not equal 360 - check directional resolution

6. P 6, line 6, why is wind data interpolated to 0.25 deg in this case? Is this a typo?

7. On p 6 there is repeated us if increased resolution while referring to reduce grid size, this should be more clear.

8. In Table 2 use consistent units for spatial resolution

9. P 6 line 22, state extreme event is in September.

10. P 7 line 5 and subsequent – use of a strange reference to calm conditions – does this mean zero wind or less than a threshold? It seems superfluous to state the models give the same result when there is no wind! Presumably there can be residual swell. Does the model use the same boundary forcing for each case? This seems implied by p 5 line 15. Are all the wind fields consistent near the open boundary? Could this be an issue – discuss.

11. P 10 line 6 this sentence is not necessarily self-evident, why?

12. P 11, lines 9-13 – the term 'period' of the peak is a bit confusing, maybe use 'duration or persistence.

13. P 11 lines 23-24, stating the obvious?

14. P 16, lines 9-10 – using coupled model not demonstrated here – don't state this unless using reference to other work

15. Watch out for missing spaces between words in a few instances e.g. p5, p11

---

## Referee Comment (RC2) · Anonymous Referee #2 · 5 Sep 2018

General Comments The paper deals with interesting topic of the performance of WAM model in the region of North and Baltic Seas. The authors analyses the sensitivity and the accuracy of the model WAM using different wind models (and considering different time steps and resolutions). They found that the time resolution is crucial (hourly data) in this region in order to reproduce the significant peak of the wave height and its correct location. Moreover, they describe which is the best (most accurate) wind forcing model in the region. The performance of the newly Sentinel-3A data is compared with other operational satellites (Jason-2 and Cryosat-2) showing the better results for Sentinel-3A in the coastal areas (mainly in the first 10km). The analysis also reveals that the Sentinel-3A quality is independent of the satellite flights direction, metocean conditions, and even for the wind direction relative to the flight direction. To

my knowledge, this is the first study that describes at this detail this kind of process and Sentinel performance in this region. In general, the manuscript is well organized and written, allowing an easy reading and following up the discussion and conclusions. In my opinion, a set of few changes would allow the publication of the manuscript in the journal. I recommend minor revision.

However, still some points that the authors should consider. 1) Are some of the wind models used for the analysis already assimilating data? If this is true, it should be considered when comparing all the results. Moreover, the validation of the winds should not include the assimilated data. 2) About EOF. I am not sure that it gives more information than observed in Figure 4. By the way, looking at Figure 4 it is clearly observed the displacement of the location of maximum Hs to the NE in the 6h models. So basically, the first mode of the EOF as you have it now are related with this "displacement". It would be nice to apply the EOF only to the 6h (or 1h) and not only for the event of 29th September, but also for the entire simulation. In that sense you can describe how are the modes related to both the time and the space variability. 3) Could you define the statistics used? RMS, Bias and correlation are well known. But what about SI? Are you referring to the RMSE normalized by the mean observed values? 4) While comparing the same model with different time resolutions seems appropriate, I do not have the same feeling when the spatial resolution comparison is done. You are comparing different models, not the same model with different resolutions. In this case, if you want to check the sensitivity of the model to different wind forcing resolutions I would suggest to do some spatial subsetting from the finer resolution.

Specific Comments P2L10-L12: This sentence seem to indicate that the wave models always worked good and with high accuracy, and bad or good results are only dependent on the meteorological models. In my opinion, it should be rephrased.

P2L18: define acronym SWAN

P3L19: what it this mail referred to?

[Figure]

P4: some acronyms not specified: i.e. JCOMM, ORBCOMM, GOES and WMO

P5L14-L15: some information about the coarser model simulation? Spatial, frequency and temporal resolutions?

P6L3-L11: there is a kind of a mess in these lines. All the models are "interpolated" to different resolutions (finner) than the original. However, the operational ECMWF of 9km (0.08°) is interpolated to 0.125°. Aren't you losing some information in this process?

Table 2 :are the models list following some order? I would ordinate them from coarser to finer (and homogenize horizontal dimension units).

P7L30-L32 : where have you seen this? is there any figure I am missing?

P8L15: change "time step" for "instant" or "event"

P8L18: I am not sure if the word ensemble here is the most appropriate. Probably you can use: numerical tests.

P10L11-L14: could you mark the in situ measurements used here in the Figure 1?

P11_L1-L3: why don't you analyze the event two days before? It is not 7m of Hs, but more than 4m.

P12-L14: do you take into account some kind of land-mask? For example, what happens if your mooring is inside a bay, and the satellite track is outside (with land between them)?

Table 3: number of data used for the analysis please.

Figure 1: explain the black and white boxes.

Figure 2: what is M and R in the text boxes (also in Figure 3)? Do the 6h and 1h models have the same number of "entries"?

Figure 4: the OBCs seems a bit different in some cases (4a and b shows wave heights in the NW corner not observed in the others. . .). Is this related only to wind forcing?

---

## Short Comment (SC1) · 24 Sep 2018

General Comments: In this work, first the authors present the validation, again in situ observations, of the results of the WAM wave model forced with different wind models with different spatial and temporal resolution for the North and Baltic Seas. Concluding that in this area the results of the WAM model are more precise when is forced with winds of lower temporal resolution. For later, using the previous model results, in order to demonstrate that the wind and wave satellite observations of the Sentinel 3 are more accurate near the coast than any other of the previous available satellite observations.

However, there are some points that in my opinion the authors should consider:

- The biggest effort of the work is in the validation of the results of the WAM wave

model, which is not reflected in the title or in the abstract of the work.

- On the other hand, in this validation different wind models are used, which are not sufficiently described and their differences and similarities are not listed (Example, if they incorporate or not assimilation of data, etc.), which is very important at the time of understand the differences in the subsequent wave results.

- Finally, mention the need to incorporate in the text the definition of the error statistics that have been used throughout the work (eg, SI)

I recommend minor revision before acceptance for publication.

Detailed comments:

1. The title could be "Synergy between WAM model simulation and Sentinel 3 observations during a extreme events"

2. With the same idea above, the abstract should be rewritten.

3. P5 Figure 1: Explain in text of the figure the three colors boxes

4. P5 line 11: There is an error in the directional resolution described

5. P6 table 2: I recommend use the same units for the spatial resolution of the different meteo models. The same along the text.

6. P8 lines 9-18: Change "ensemble" for "different model experiments" or "numerical tests"

7. P11 lines 9-13: The concept of "period of the peak" can be confusing in this context of results of wave models and it is more frequent to call it "storm duration"

8. P16, lines 9-10: Explain why do you think that using wave-atmosphere coupling models could you improved your results

---

## Editor Comment (EC1) · A. Sánchez-Arcilla (Editor) · 3 Oct 2018

Based on the performed reviews this paper should emphasize what is actually novel and that is the satellite data assessment using a wave model. In a comparison the difference between wind fields as driver for the resulting wave fields should be analysed in terms of objective parameters. For instance running the meteorological model or sampling a given wind field at different grid sizes. This objective comparison trying to modify only one element at a time within the whole comparison process would add value to the paper and increase the strength of the essential message which should be related to the quality of satellite data and its limitations.

The robustness of the Sentinel-3A data should be emphasized, stressing the impor-

tance of data quality regardless of flight direction or wind direction.

Finally the title should be reconsidered since it does not reflect the wave model validation effort which is one of the main components of the paper.

---

## Author Comment (AC1) · 17 Oct 2018

Overview:

This paper focusses on the issue of improving wave model performance by improving the driving wind fields. It is following many other papers on this topic but does add some originality in its assessment of the differences between wave model outputs. The wave model outputs are compared with in situ data as the ground truth. The results are similar to previous work in demonstrating that wave model results are sensitive to the wind forcing, which is not surprising or very new. A secondary part of the work is in examining the quality of satellite data on significant wave height, showing that the most recent satellite Sentinel-3A is superior to previous satellites, especially within 10km of the coast, but still requires bias correction of the significant wave height before any improvement in wave model performance could be achieved by its assimilation into an operational model.

Authors: We would like to thank to the reviewers for the valuable comments about our manuscript. Important points have been raised that helped us to improve the clarity and the understanding of our study.

General:

The paper is generally clearly written and readable, apart from a few errors (listed below). However it could be improved by some reorganisation and rationalisation.

Authors: We have revised our manuscript – reorganizing it following reviewers' suggestions.

Authors: The revised manuscript reflects these comments and a point-to-point response to them is provided below.

The motivation for the study and the order of the presentation could be sharpened up.

Authors: This has been modified in the revised manuscript

The satellite data assessment is stated to be the main motivation but the bulk of the paper addresses the wave model forcing by different atmospheric model wind fields and assessing which performs better. This exercise is somewhat flawed as the wind fields come from different models and it is not clear what are the differences between these. This should be discussed further.

Authors: We agree. The first part of the paper is focused on the wave model forcing by different atmospheric model wind fields and assessing which performs better. This has been now stressed in the introduction and the abstract has been also modified, accordingly. The differences between the different models is discussed in more detail.

Some are reanalysis products and some are forecasts. Presumably these have different wind datasets assimilated into them. Thus it is not really a like-for-like comparison. The issue of model spatial and temporal resolution could be more rigorously assessed by sampling the same wind field at different resolutions, but this is not done.

Authors: We agree with the comment. We don't aim here to make assessment and quantitative analyses of the different wind forcing provided by different centers and sources. The wind data sets indeed differ in their horizontal and spatial resolution as well as the data that are assimilated (or not). This is further discussed in Section 2.3. Our motivation is explained now better in the introduction. The aim in this study is to give an overview of the performance of the wave model over the considered area. These analyses were needed to define our "reference forcing" and further proceed with those model simulations and assessments of the satellite data of Sentinel-3A in comparisons with older altimeter data widely used for validations, in-situ observations together with the model simulations. This has been also emphasized in the Discussion.

The authors state that the WAM model performs well with all the datasets, which has already been demonstrated.

Authors: The only publication we are aware of comparing wave data with ERA-interim and ERA5 wind forcing is Nose et al. (2018) "Predictability of storm wave heights in the ice-free Beaufort Sea". This is, though, focused for a different region and for only two months. This has now been referred to in the text and added to the references. Since wind re-analyses of ERA-5 are still new there are still no sufficient publications about WAM performance under this forcing on regional scales.

It is also known that higher spatial and temporal resolution improves the wave model performance.

Authors: We agree. More discussion and references have been added. We demonstrate in our study the wave model performance for the considered area with the available wind forcing. This has been re-formulated now to make our statements more clear.

There is an interesting comparison between results, using an EOF analysis to show differences in wind speed, direction and location of the maximum affecting the maximum wave height in an extreme event. Using reference to a model ensemble where it is demonstrated that some wind fields are better quality than others seem somewhat perverse – can this be justified?

Authors: In our study we do not aim at stating that one wind field is better than the other. Rather we wanted to demonstrate with which wind field WAM produces the best results in order to compare the satellite data with the model. We rephrased this and made our statements clearer.

Overall this paper needs moderate revision before acceptance for publication.

Detailed comments:

1. The title should include 'of wind and waves' after 'simulations'

Authors: Thanks! It has been added to the title.

2. The first line of the abstract is somewhat misleading as only satellite wave data are evaluated. Also this part of the work is second to the study of the spatial and temporal resolution of the wind forcing for the wave model and so perhaps should not be introduced first.

Authors: We agree. The abstract has be re-organized and rephrased following this comment.

3. P 2 line 7, suggest inserting 'in determining' to replace 'of'

Authors: This has been inserted.

4. P 2 line 31 'flagging of (data)' – explain further, this is rather cryptic.

Authors: The word has been replaced by discarding.

5. P 5 line 11 – 30 times 15 does not equal 360 - check directional resolution

Authors: You are right. It is 24 directions.

6. P 6, line 6, why is wind data interpolated to 0.25 deg in this case? Is this a typo?

Authors: This is how the output is made available. We rephrased the paragraph.

7. On p 6 there is repeated us if increased resolution while referring to reduce grid size, this should be more clear.

Authors: This has been modified.

8. In Table 2 use consistent units for spatial resolution

Authors: Thank you. The units have been homogenized.

9. P 6 line 22, state extreme event is in September.

Authors: This has been added.

10. P 7 line 5 and subsequent – use of a strange reference to calm conditions – does this mean zero wind or less than a threshold? It seems superfluous to state the models give the same result when there is no wind!

Authors: Thank you. The word calm was not correctly used here. What we meant were wind speeds below 5 Bft or significant wave height below 2m. Therefore, we modified that.

11. Presumably there can be residual swell. Does the model use the same boundary forcing for each case? This seems implied by p 5 line 15. Are all the wind fields consistent near the open boundary? Could this be an issue – discuss.

Authors: Yes, you are right. In our study the boundary conditions are the same for all the model simulations. This has been added and discussed in the revised manuscript.

12. P 10 line 6 this sentence is not necessarily self-evident, why?

Authors: We agree. The sentence has been rephrased and we made our statements clearer.

13. P 11, lines 9-13 – the term 'period' of the peak is a bit confusing, maybe use 'duration or persistence.

Authors: This has been changed.

14. P 11 lines 23-24, stating the obvious?

Authors: We agree. However we decided to keep these statements since they summarize the importance of our findings in the first part of the manuscript and provide a transition to the second part that is on the synergy analyses. We provided arguments for choosing the wave-model run for validation analyses. This is now clarified.

15. P 16, lines 9-10 – using coupled model not demonstrated here – don't state this unless using reference to other work

Authors: Our apologies. This has been deleted.

16. Watch out for missing spaces between words in a few instances e.g. p5, p11

Authors: Thanks and we are very sorry for this. It has been fixed.

---

## Author Comment (AC2) · 17 Oct 2018

General Comments The paper deals with interesting topic of the performance of WAM model in the region of North and Baltic Seas. The authors analyses the sensitivity and the accuracy of the model WAM using different wind models (and considering different time steps and resolutions). They found that the time resolution is crucial (hourly data) in this region in order to reproduce the significant peak of the wave height and its correct location. Moreover, they describe which is the best (most accurate) wind forcing model in the region. The performance of the newly Sentinel-3A data is compared with other operational satellites (Jason-2 and Cryosat-2) showing the better results for Sentinel-3A in the coastal areas (mainly in the first 10km). The analysis also reveals that the Sentinel-3A quality is independent of the satellite flights direction, metocean conditions, and even for the wind direction relative to the flight direction. To my knowledge, this is the first study that describes at this detail this kind of process and Sentinel performance in this region. In general, the manuscript is well organized and written, allowing an easy reading and following up the discussion and conclusions. In my opinion, a set of few changes would allow the publication of the manuscript in the journal. I recommend minor revision.

Authors: We are very grateful for the kind considerations about our manuscript

However, still some points that the authors should consider.

 1) Are some of the wind models used for the analysis already assimilating data? If this is true, it should be considered when comparing all the results. Moreover, the validation of the winds should not include the assimilated data.

Authors:  We agree and this is now discussed in the revised manuscript. For the ECMWF reanalysis, near the surface, in-situ wind data were indeed part of the data provided to the 4Dvar. Sure the wind data enter in the system but so are all the other data and the vibrational approach will attempt to improve the fit with all these observations but it does not mean that one can expect a perfect fit. Actually, it is one of the key diagnostics when looking at the performance of the DA system, namely the fit to the data from the first guess and the analysis. Generally one expects to have a better fit with the analysis. So it is still worthwhile to look at the comparison.
Also, the short range forecasts by the ECMWF from the 4Dvar system have been influenced by the data assimilation because the assimilation is performed over a 6 or 12 window with data that can be more recent (by a few hours) than the start time of each forecast. For the DWD forecast, the in-situ wind data is assimilated into the analysis used to initialize the forecast but for the forecast itself, no data was assimilated.
For the coastDat-3 data set no data was assimilated at all.
This explanation has been added in Section 2.3.

2) About EOF. I am not sure that it gives more information than observed in Figure 4. By the way, looking at Figure 4 it is clearly observed the displacement of the location of maximum Hs to the NE in the 6h models. So basically, the first mode of the EOF as you have it now are related with this "displacement". It would be nice to apply the EOF only to the 6h (or 1h) and not only for the event of 29th September, but also for the entire simulation. In that sense you can describe how are the modes related to both the time and the space variability.

Authors: The EOF analysis is carried out for one event for the whole ensemble in order to estimate the differences between the model simulations with the different wind forcings. Sure, most of the differences can already be observed by looking at each ensemble member individually but this analysis combines all of them. We now did the EOF analysis for June till August 2016 for the difference between the two ERA5 model simulations in order to estimate the influence of the different temporal resolutions over time, but this only revealed again, that during normal conditions both model simulations are very similar with an explained variance of only 3.13% for the dominant mode of the EOF and even less

for further modes (Figure 1). The large differences between the model simulations appear only for extreme events, which is why we concentrated on that. We have now added the above explanations in the revised manuscript.

[Figure]

Figure 1: The explained variance for each mode of the EOF analysis for the difference between the model simulation with hourly and six hourly ERA5 wind forcing for summer 2016.

3) Could you define the statistics used? RMS, Bias and correlation are well known. But what about SI? Are you referring to the RMSE normalized by the mean observed values?

Authors: We agree. The statistical analyses that have been used in our study are have been defined in the Appendix.

4) While comparing the same model with different time resolutions seems appropriate, I do not have the same feeling when the spatial resolution comparison is done. You are comparing different models, not the same model with different resolutions. In this case, if you want to check the sensitivity of the model to different wind forcing resolutions I would suggest to do some spatial subsetting from the finer resolution.

Authors: Thank you for the comment. The wind data sets that have been used here indeed differ in their horizontal and spatial resolution as well as the data that are assimilated (or not). This is further discussed in Section 2.3. The aim in this study is to give an overview of the performance of the wave model over the considered area with different available wind products, so that other scientist planning to performed similar simulations have an idea what to expect. Our motivation is explained now clearer in the introduction. These comparisons were also used do define our "reference forcing" and further proceed with those model simulations and assessments of the satellite data of Sentinel-3A in comparisons with older altimeter data widely used for validations, in-situ observations together with the model simulations. This has been also emphasized in the Discussion.

Specific Comments

P2L10-L12: This sentence seem to indicate that the wave models always worked good and with high accuracy, and bad or good results are only dependent on the meteorological models. In my opinion, it should be rephrased.

Authors: In our manuscript, we refer to the paper by Cavaleri and Bertotti, 1997 and the state-of-the- art provided by that manuscript.

"Our experience as wave modellers, hence of users of surface wind products, strongly suggests the lack of sufficient resolution as a likely culprit. As pointed out in the introduction, advanced wave models are at present more accurate than the meteorological models, and they are therefore good indicators of the quality of the driving wind fields."
We have now re-written this in the revised manuscript making this clearer.

P2L18: define acronym SWAN

Authors: We have added the full name.

P3L19: what it this mail referred to?

Authors: We have changed the source of the data.

P4: some acronyms not specified: i.e. JCOMM, ORBCOMM, GOES and WMO

Authors: This has been fixed in the revised manuscript:

P5L14-L15: some information about the coarser model simulation? Spatial, frequency and temporal resolutions?

Authors: The information has been added.

P6L3-L11: there is a kind of a mess in these lines. All the models are "interpolated" to different resolutions (finner) than the original. However, the operational ECMWF of 9km (0.08 ∘ ) is interpolated to 0.125 ∘ . Aren't you losing some information in this process?

Authors: Our apologies for the confusion. This has been re-written in the revised manuscript. Explaining that this is how the model output is made available. We have added also more information about the wind forcing data.

Table 2: are the models list following some order? I would ordinate them from coarser to finer (and homogenize horizontal dimension units).

Authors: We are sorry for the confusion. The units have been homogenized and ordinated.

P7L30-L32 : where have you seen this? is there any figure I am missing?

Authors: This is clarified in the revised manuscript. We calculated the bias and haven't shown any figure for this.

P8L15: change "time step" for "instant" or "event"

Authors: We agree. Thank you. This has been changed.

P8L18: I am not sure if the word ensemble here is the most appropriate. Probably you can use: numerical tests.

Authors: We decided to stick to the word "ensemble" here, because it better reflects our idea behind using the different model experiments.

P10L11-L14: could you mark the in situ measurements used here in the Figure 1?

Authors: They are marked with the white box. We rephrased the sentence to make this clear.

P11_L1-L3: why don't you analyze the event two days before? It is not 7m of Hs, but more than 4m.

Authors: The extreme event is analyzed as well in Section 3.2.3. We refer to the section and corresponding values.

P12-L14: do you take into account some kind of land-mask? For example, what happens if your mooring is inside a bay, and the satellite track is outside (with land between them)?

Authors: You are right. This has not been taken into account. We clarified this in the revised manuscript.

Table 3: number of data used for the analysis please.

Authors: This has been added.

Figure 1: explain the black and white boxes.

Authors: We have added an explanation in the figure caption.

Figure 2: what is M and R in the text boxes (also in Figure 3)?

Authors: We have added an explanation in the figure caption.

Do the 6h and 1h models have the same number of "entries"?

Authors: Yes, because only the wind forcing is 6 hourly, but the wave model still has hourly output.

Figure 4: the OBCs seems a bit different in some cases (4a and b shows wave heights in the NW corner not observed in the others…). Is this related only to wind forcing?

Authors: The OBCs are the same for all model simulations. The differences are solely because of the different wind forcings. We have now made this clear in the revised manuscript.

---

## Author Comment (AC3) · 17 Oct 2018

General Comments: In this work, first the authors present the validation, again in situ observations, of the results of the WAM wave model forced with different wind models with different spatial and temporal resolution for the North and Baltic Seas. Concluding that in this area the results of the WAM model are more precise when is forced with winds of lower temporal resolution. For later, using the previous model results, in order to demonstrate that the wind and wave satellite observations of the Sentinel 3 are more accurate near the coast than any other of the previous available satellite observations. However, there are some points that in my opinion the authors should consider:

- The biggest effort of the work is in the validation of the results of the WAM wave model, which is not reflected in the title or in the abstract of the work.

Authors: We agree. The title has been changed now to include also the WAM wave model validations.

- On the other hand, in this validation different wind models are used, which are not sufficiently described and their differences and similarities are not listed (Example, if they incorporate or not assimilation of data, etc.), which is very important at the time of understand the differences in the subsequent wave results.

Authors: This is a very good point also made by one of the other reviewers. We added in the revised version more information about the different wind data used in this study.

- Finally, mention the need to incorporate in the text the definition of the error statistics that have been used throughout the work (eg, SI)

Authors: Thank you for this comment – this is very good point. This has been added now.

I recommend minor revision before acceptance for publication.

Detailed comments:

1. The title could be "Synergy between WAM model simulation and Sentinel 3 observations during a extreme events"

Authors: Thank you for the suggestion. We agree that it is important to add the wave model. We however decided not to specify the name of the wave model WAM in the title but just to use wave, in order to attract a wider audience to read our study.

The title now is "Synergy of wind wave model simulations and satellite observations during extreme events"

2. With the same idea above, the abstract should be rewritten.

Authors: You are right. We have now re-written the abstract incorporating also the information about the wave model WAM simulations.

3. P5 Figure 1: Explain in text of the figure the three colors boxes

Authors: We agree. This has been added.

4. P5 line 11: There is an error in the directional resolution described

Authors: Thank you and apologies for our mistake. This has been corrected.

5. P6 table 2: I recommend use the same units for the spatial resolution of the different meteo models. The same along the text.

Authors: We agree. The units have been homogenized.

6. P8 lines 9-18: Change "ensemble" for "different model experiments" or "numerical tests"

Authors: We decided to use "ensemble" here, because it better reflects our idea behind using the different model experiments.

7. P11 lines 9-13: The concept of "period of the peak" can be confusing in this context of results of wave models and it is more frequent to call it "storm duration"

Authors: "Period of the peak" has been changed to "duration of the peak".

8. P16, lines 9-10: Explain why do you think that using wave-atmosphere coupling models could you improved your results

Authors: This statement has been modified in the revised manuscript.

---

## Author Comment (AC4) · 17 Oct 2018

**A. Sánchez-Arcilla (Editor)** agustin.arcilla@upc.edu

Based on the performed reviews this paper should emphasize what is actually novel and that is the satellite data assessment using a wave model. In a comparison the difference between wind fields as driver for the resulting wave fields should be analysed in terms of objective parameters. For instance running the meteorological model or sampling a given wind field at different grid sizes. This objective comparison trying to modify only one element at a time within the whole comparison process would add value to the paper and increase the strength of the essential message which should be related to the quality of satellite data and its limitations.

Dear Editor,

We would like to thank the three reviewers for their valuable comments about our manuscript. They have raised important points that helped us to improve the clarity and the understanding of the study. The revised manuscript reflects these comments and a point-to-point response to them is provided below. The changes can be checked in the track changes document. We have now revised the manuscript and emphasized what is the novel. In particular, we have now made our manuscript much clearer, following the reviewers' suggestions and added more analyses and discussion of the different wind data used. Also more detailed analyses on the comparisons have been made and the limitations of the satellite data and methods used have been identified.

The robustness of the Sentinel-3A data should be emphasized, stressing the importance of data quality regardless of flight direction or wind direction.

Authors: Thanks for this comment, which we believe is important for our work. As mentioned above we have rewritten the statement emphasizing on the importance of the new Sentinel-3A data quality regardless of flight direction or wind direction.

Finally the title should be reconsidered since it does not reflect the wave model validation effort which is one of the main components of the paper.

Authors: Following also the comments of two reviewers, the title of the revised manuscript has been modified, accordingly.

We hope that this submission answers the comments and questions of the reviewers.